# The need for blood transfusion therapy is associated with increased mortality in children with traumatic brain injury

Madhuradhar Chegondi[1,2]*, Jose F. Hernandez Rivera[3], Fuad Alkhoury[4], Balagangadhar R. Totapally[5,6]

1 Division of Critical Care Medicine, Stead Family Children's Hospital, Iowa City, Iowa, United States of America, 2 Department of Pediatrics, Carver College of Medicine, University of Iowa, Iowa City, Iowa, United States of America, 3 Division of Critical Care Medicine, Children's Hospital, University of Florida, Gainesville, Florida, United States of America, 4 Department of Pediatric Surgery, Nicklaus Children's Hospital, Miami, Florida, United States of America, 5 Division of Critical Care Medicine, Nicklaus Children's Hospital, Miami, Florida, United States of America, 6 Herbert Wertheim College of Medicine, Florida International University, Miami, Florida, United States of America

* madhuradhar-chegondi@uiowa.edu

**Data Availability Statement:** The data is owned by the Healthcare Cost and Utilization Project (HCUP), which is maintained by the Agency for Healthcare Research and Quality (AHRQ). The data is available

## Abstract

### Objective

Blood transfusion therapy (BTT) is widely used in trauma patients. However, the adverse effects of BTT in pediatric trauma patients with traumatic brain injury (TBI) were poorly studied. The objective of this study is to evaluate the effect of BTT on mortality in children with severe TBI.

### Methods

In this retrospective cohort analysis, we analyzed 2012 and 2016 Kids' Inpatient Databases and used a weighted sample to obtain national outcome estimates. We included children aged 1 month to 21 years with TBI who were mechanically ventilated, considered severe TBI; we then compared the demographics, comorbidities, and mortality rates of those patients who had undergone BTT to those who did not. Statistical analysis was performed using the chi-squared test and regression models. In addition, in a correlative propensity score matched analysis, cases (BTT) were matched 1:1 with controls (non-BTT) based on age, gender, hospital region, income quartiles, race, and All Patients Refined Diagnosis Related Groups (APRDRG) severity of illness scores to minimize the effect of confounding variables between the groups.

### Results

Out of 87,980 children with a diagnosis of TBI, 17,199 (19.5%) with severe TBI were included in the analysis. BTT was documented in 3184 (18.5%) children. Among BTT group, the mortality was higher compared to non-BTT group [31.6% (29.7–33.5%) vs. 14.4 (13.7–15.1%), (OR 2.2, 95% CI 1.9–2.6; p<0.05)]. In the BTT group, infants and adolescents, white race, APRDRG severity of illness, cardiac arrest, platelet, and coagulation

to the public by paying a licensing fee and signing a data use agreement. The authors data use agreement doesn't allow them to share the data directly. Although this proprietary data can be used freely without any restrictions for research purpose, to avoid commercial and competitive usage of the data there are restrictions on data sharing in this Data Use Agreement. Please reach out to AHRQ HCUP (www.hcup-us.ahrq.gov) for more information on accessing HCUP data.

**Funding:** The author(s) received no specific funding for this work.

**Competing interests:** The authors have declared that no competing interests exist.

factor transfusions were associated with higher mortality. In a propensity-matched analysis, BTT associated with a higher risk of mortality (32.1% [30.1–34.2] vs. 17.4% [15.8–19.1], p<0.05; OR: 2.2, 95% CI: 1.9–2.6).

## Conclusion

In children with severe TBI, blood transfusion therapy is associated with higher mortality.

## Introduction

Traumatic brain injury (TBI) in children is a major public health problem globally, with the incidence ranging from 47 to 280 per 100,000 children [1]. However, in the United States, the estimated annual incidence of TBI is 691 per 100,000 children [2]. TBI is one of the most common causes of trauma-related deaths in children. The mortality from pediatric TBI increased recently from 4.42 to 5.17 per 100000 children [3]. In the United States, children aged 0–4 years and adolescents aged 15–19 years are most likely to have a TBI-related emergency department visit or be hospitalized for a TBI compared to other children of different age groups [2].

In children, TBI results in significant healthcare resource utilization and requires multiple interventions [4]. Blood transfusion therapy (BTT) is often used in pediatric TBI. One of the crucial aspects of TBI management is preventing secondary insults to the brain by avoiding hypotension, hypoxia, and anemia [5]. However, there is no optimal threshold for the transfusion of packed red blood cells (PRBC) and other blood components in pediatric trauma and TBI [6]. Traditionally, PRBCs were transfused to keep hemoglobin levels above 10gm/dL to maintain oxygen-carrying capacity [5, 6]. Recent evidence supports restrictive transfusion strategies in critically ill children, including TBI, withholding transfusion for hemoglobin levels above 7 gm/dL [7, 8]. BTT was found to have higher adverse events in children compared to adults and could be life-threatening [9]. The deleterious effects of BTT have been poorly studied in pediatric trauma patients with TBI. The objective of this study was to evaluate the effect of BTT on mortality in children with severe TBI using a national pediatric hospital discharge database.

## Materials and methods

This study was exempted from the institutional review board (IRB) review and conducted as per Healthcare Cost and Utilization Project (HCUP) guidelines. We performed a retrospective review of the Kid's Inpatient Database (KID) 2012 and 2016 for all reported TBI cases defined by the International Classification of Diseases, 9th and 10th Edition (ICD-9, ICD-10) diagnostic codes 800.XX, 801.XX, 803.XX, 804.XX, 850.XX, 851.XX, 852.XX, 853.XX, and 854.XX and ICD-10 diagnostic codes S02.0, S02.1, S06.0, S06.1, S06.2, S03.30, S03.31, S03.32, S03.33, S09.x. The ICD-9 and ICD-10 procedure codes, V58.2, and Z51.89, respectively, were used to identify children who received BTT. Mechanically ventilated patients were identified using Clinical Classification Software for Procedures (CCS-PCS) code 216. ICD diagnosis and procedures codes and the Clinical Classification Software Diagnoses (CCS) and CCS-PCS codes were used to identify various other variables, including platelet and coagulation factor transfusions. The CCS diagnosis and CCS-PCS codes identify diagnostic and procedure groups, respectively. The CCS and CCS-PCS were created by HCUP to merge multiple individual ICD-9 and 10 diagnosis and procedure codes into a smaller number of clinically meaningful categories for a

more uniform and standardized coding system [10]. We included patients who were discharged from hospitals with a diagnosis of TBI with ages between 1 month to less than 21 years who were mechanically ventilated. As the KID database does not include Glasgow coma scale (GCS) scores, we used mechanical ventilation as a surrogate marker of severe TBI. We defined internal injury as thoracic and abdominal injury resulting from blunt trauma.

Cases with a discharge disposition to the short-term hospital were excluded to avoid double counting.

The KID is one in a family of databases and software tools developed as part of the HCUP. The KID contains hospital discharge information from U.S. hospitals that participate in HCUP. Participating institutions are defined as short-term, non-federal, general, and specialty hospitals, excluding rehabilitation hospitals. It was explicitly designed to permit researchers to study a broad range of conditions and procedures related to child health issues. The KID contains clinical and resource use information included in a typical discharge abstract, with safeguards to protect the privacy of individual patients, physicians, and hospitals (as required by data sources). Researchers and policymakers can use the KID to identify, track, and analyze national trends in healthcare utilization, access, charges, quality, and outcomes. The KID excludes data elements that could directly or indirectly identify individuals. Normal newborns were sampled at a rate of 10%, while complicated newborns and other pediatric discharges (age 20 years or less at the time of admission) were sampled at a rate of 80%. Sampling weight is provided for each record so that national; estimates can be calculated. The KID for 2012 includes data from 4,179 hospitals and 4200 hospitals during 2016 [10]. The All-Patient Refined Diagnosis Related Groups (APR-DRG) are patient classification methods to describe a hospital's case mix with the severity of illness and its incurring costs [10].

## Statistical analysis

The prevalence of TBI is reported per 10,000 discharges. The mortality rate is reported per 100 children with TBI. All categorical variables (gender, race, patient location, median household income (MHI) were analyzed using the chi-square test. HCUP defines MHI by the zip code in which the child resides. The zip codes are stratified by quartiles, with quartile 1 representing the lowest and quartile 4 representing the highest median income. Binomial data were presented as odds ratios with 95% confidence intervals and p-values. All continuous variables (total charges, length of stay, number of procedures, and number of diagnoses) were compared using the Mann-Whitney U test. Their data is presented as the median and interquartile range (IQR). P-values <0.05 were considered statistically significant. All data were weighted according to HCUP recommendations before analysis to calculate national estimates.

Variables associated with increased mortality in children with severe TBI were evaluated using univariate analysis. Variables significantly associated with mortality at a p-value <0.1 were then included in a regression model. A binary logistic regression was performed to ascertain the effects of race/ethnicity, age groups, median household income quartile of patient's zip code, transfer-in status, payor status, APR-DRG severity of illness score, thoracic and abdominal injury, nonaccidental trauma (NAT), cardiopulmonary resuscitation (CPR)/cardiac arrest, treatment in a children's hospital, receiving PRBC (BTT), platelets or coagulation factor transfusions on the likelihood of mortality.

Propensity score-matched analysis was done to evaluate the effect of BTT on mortality. Cases (BTT) were matched 1:1 with controls (non-BTT) using propensity score based on the variables: age groups, gender, hospital region, income quartiles, race/ethnicity, admission to children's hospital, transfer-in status, year of admission, thoracic and abdominal injury, non-accidental injury, traumatic shock, sepsis, cardiac arrest or cardiopulmonary resuscitation,

platelet, and coagulation factor transfusion, and APR-DRG severity of illness score. A fuzzy match at an 80% match level was allowed for propensity matching. SPSS version 26.0 (IBM Corporation, Armonk, NY) and StatCalc of Epi Info™ (Centers for Disease Control and Prevention, Atlanta) were used for the analysis.

## Results

### Demographics

A total of 87,980 children aged 1 month to less than 21 years trauma patients with a diagnosis of TBI were discharged during 2012 and 2016; 17,199 (19.6%) received mechanical ventilation, considered as severe TBI, and were included in the analysis. BTT was recorded in 3,184 (18.5%) children with severe TBI. The demographic characteristics in the BTT group and non-BTT groups are presented in Table 1. Females were more likely to receive BTT (OR:1.4, 95% CI: 1.3–1.5; p<0.001). The median age group was lower in the BTT group, 10.8 (10.5–11.1) years, compared to the non-BTT group, 13.2 (13.1–13.3) years (p<0.001). White children and children with Government insurance were more likely to receive BTT (Table 1).

### Clinical characteristics

Clinical characteristics and interventions in BTT and non-BTT groups are presented in Table 2. In the BTT group, the thoracic and abdominal injury was documented in about half

**Table 1. Differences in demographic variables among children with traumatic brain injury and mechanical ventilation with and without blood transfusion therapy.** The values are presented as percentages (95% Confidence intervals).

| Variable | Blood transfusion group % (95%, CI) (n = 3184) | No Blood transfusion Group % (95%, CI) (n = 14014) | OR (CI); p-value |
|---|---|---|---|
| *Gender* | | | |
| Female | 36.5 (34.6–38.5) | 29.2 (28.4–30.1) | 1.4 (1.3–1.5); <0.001 |
| *Race/Ethnicity* | | | |
| White | 49.2 (47–51.4) | 55.8 (54.8–56.9) | <0.001 |
| Black | 20.1 (18.4–21.9) | 17.1 (16.4–17.9) | |
| Hispanic | 20.4 (18.7–22.3) | 18.8 (18–19.6) | |
| Other | 10.3 (9–11.6) | 8.2 (7.7–8.8) | |
| *Age groups* | | | |
| Infants | 16.3 (14.8–17.8) | 6 (5.5–6.4) | <0.001 |
| Toddlers | 10.9 (9.7–12.3) | 7.4 (6.9–7.9) | |
| Early childhood | 6.7 (5.7–7.8) | 6.6 (6.1–7.1) | |
| Middle childhood | 10.5 (9.3–11.8) | 10.8 (10.2–11.5) | |
| Early adolescence | 34.6 (32.7–36.6) | 42.4 (41.5–43.4) | |
| Late adolescence | 21 (19.4–22.7) | 26.8 (25.9–27.6) | |
| *Region* | | | |
| Northeast | 12.9 (11.6–14.3) | 12.9 (12.2–13.5) | <0.001 |
| Midwest | 21.8 (20.2–23.5) | 23.4 (22.6–24.2) | |
| South | 45.4 (43.4–47.5) | 39.8 (38.9–40.8) | |
| West | 19.9 (18.3–21.6) | 23.9 (23.1–24.7) | |
| *Weekend admission* | 33.6 (31.7–35.5) | 35.4 (34.5–36.3) | 0.9 (0.8–1); 0.05 |
| *Insurance payer* | | | |
| Government | 43.4 (41.4–45.4) | 40.4 (39.4–41.4) | <0.001 |
| Private | 41.9 (39.9–43.9) | 45.9 (44.9–46.9) | |
| Others | 14.7 (13.3–16.2) | 13.7 (13–14.4) | |
| *Children's Hospital* | 21.6 (19.9–23.4) | 17.5 (16.7–18.3) | NR |

**Table 2. Clinical characteristics of children with traumatic brain injury and mechanical ventilation with and without blood transfusion therapy.** The values are presented as percentages (95% Confidence intervals).

| Variable | Blood transfusion group % (95%, CI) (n = 3,184) | No Blood transfusion group % (95%, CI) (n = 14,014) | OR (CI); p-value |
|---|---|---|---|
| Thoracic and abdominal injury | 50.6 (48.6–52.7) | 35.8 (34.9–36.8) | 1.8 (1.7–1.9); <0.001 |
| Non-accidental trauma | 11.1 (9.8–12.4) | 3.5 (3.2–3.9) | 3.4 (2.9–3.9); <0.001 |
| Septic shock | 1.9 (4–5.8) | 1.4 (3.5–4.2) | 1.3 (1–1.8); 0.04 |
| Bacterial infection | 10.6 (9.4–11.9) | 8.7 (8.1–9.2) | 1.2 (1–1.4); 0.001 |
| CPR or Cardiac arrest | 13.1 (9.8–12.4) | 5 (4–4.8) | 2.9 (2.5–3.2); <0.001 |
| Brain death | 21.1 (19.5–22.9) | 9.4 (8.9–10) | 2.6 (2.3–2.8); <0.001 |
| DNR status | 6.8 (5.8–7.9) | 3.1 (2.8–3.4) | 2.3 (1.9–2.7); <0.001 |
| *Interventions* | | | |
| MRI brain | 5.8 (4.9–6.9) | 3.9 (3.5–4.3) | 1.5 (1.3–1.8); <0.001 |
| EEG | 5.8 (4.9–6.8) | 3.5 (3.2–3.9) | 1.7 (1.4–2); <0.001 |
| Platelet transfusion | 16.8 (15.3–18.4) | 0.6 (0.5–0.8) | 33.8 (22.7–42.7); <0.001 |
| CF Transfusion | 1.5 (1–2) | 0.1 (0.1–0.2) | 16.1 (8.8–29.8); <0.001 |
| EVD | 2.7 (2.1–3.5) | 2 (1.7–2.2) | 1.4 (1–1.8); 0.01 |

CPR = Cardiopulmonary Resuscitation, CF = Coagulation Factor, CI = Confidence Interval, DNR = Do Not
Resuscitate, EEG = Electroencephalogram, EVD = External Ventricular Drain, MRI = Magnetic Resonance Imaging, OR = Odds Ratio, TBI-MV = Traumatic Brain
Injury with Mechanical Ventilation

of the patients, and nonaccidental trauma (NAT) was present in a significantly higher proportion (11.1% vs. 3.5%). The incidence of complications, such as bacterial infection, severe sepsis/shock, cardiac arrest, and brain death, were more common in the BTT group. Similarly, patients with brain death and do not resuscitate (DNR) orders were significantly higher in the BTT group. In terms of resource utilization and interventions, brain imaging, electroencephalogram (EEG), and external ventricular drain (EVD) placement were significantly higher in the BTT group (Table 2).

## Transfusions

BTT was reported in 18.5% of severe TBI cases. APR-DRG severity of illness was greater in the BTT group. Platelet and coagulation factor transfusion requirements were higher in BTT group, 16.8% vs. 0.6% and 1.5% vs 0.1%, respectively.

## Outcomes

The median LOS [15.4 (14.7–16.1) vs. 12.4 (12.1–12.7) days] and total charges [248,969 (238,742–259,197) vs. 192,393 (187330–197456) US dollars] were higher in BTT group compared to non-BTT group. The overall mortality in children with severe TBI was 17.5% (16.9–18.2%). Among BTT group, the mortality was higher compared to non-BTT group [31.6% (29.7–33.5%) vs. 14.4 (13.7–15.1%); (OR 2.2, 95% CI 1.9–2.6; p<0.05)]. There was no mortality difference between 2012 and 2016 discharges.

## Univariate analysis

On univariate analysis of all children with severe TBI, infants, toddler, adolescent age groups, and Black race were significantly associated with increased risk of mortality. There were no significant differences in mortality rate based on other demographic characteristics such as

weekday vs. weekend admission, income quartile, and hospital characteristics. The APR-DRG severity of illness was strongly associated with increased mortality. Cardiac arrest or CPR requirement, blood, platelet, and coagulation factor transfusions were also associated with higher mortality (Table 3).

Similarly, a subgroup analysis of BTT patients showed significantly different gender, race, age group distribution between survivors and non-survivors. The median ages of survivors vs. non-survivors among the BTT group were 10.3 (9.9–10.6) years vs. 11.9 (11.3–12.5) years (p<0.001).

The median hospital LOS and hospital charges were significantly higher among survivors in the BTT group, 21.1 (20.2–21.9) days vs. 3.2 (2.8–3.6) days (p<0.001) and 306,875 (293,555–320,195) USD vs. 124,477 (114650–134304) USD (p<0.001), respectively.

## Multivariable regression analysis

The logistic regression model was statistically significant, $\chi2(25) = 3270$, $p < 0.001$. The model was a good fit with Hosmer and Lemeshow test p = 0.088. The model explained 33% (Nagelkerke $R^2$) of the variance in mortality and correctly classified 86.4% of cases. The model has correctly predicted survival or death in 86.4% of cases. Binary regression analysis showed significant differences in the distribution of age groups, race/ethnicity, median household income quartiles, payor groups, APR-DRG severity of illness, admission to children's hospital, cardiac arrest or CPR event, blood, and blood product transfusions between survivor and non-survivor groups (Table 4). After adjusting for other confounding factors, BTT was significantly associated with increased mortality risk (OR:1.73; 95% CI:1.53–1.95; p<0.001).

**Propensity score matching.** In a propensity score-matched analysis, 1881 patients in the BTT group were matched 1:1 with a control group. All baseline demographic and clinical characteristics were equally distributed between the two groups. The mortality rate remained significantly higher in the BTT group compared to the matched control group (32.1% [30.1–34.2] vs. 17.4% [15.8–19.1], p<0.05; OR: 2.2, 95% CI: 1.9–2.6). The median (IQR) length of stay (11 [3–22] vs. 7.9 [3–18] days) and total hospital charges (180,601 [91,966–329,845] vs. 120,230 [55,388–261,562] US dollars) were higher for the BTT group (p<0.05).

## Discussion

In this retrospective observational study, propensity score-matched analysis of a large national database, blood transfusion therapy was significantly associated with higher mortality in children with severe TBI. Our results are similar to a single-center retrospective study which reported a significantly increased risk of death in children with TBI who received any blood component transfusion therapy [11]. In another study in adults, national data registry analysis showed a similar increased mortality risk [1.2 (95% CI 1.1–1.3)] with blood transfusion therapy [12].

After adjusting for variables affecting mortality, blood transfusion was associated with 1.7 times the risk of mortality compared to not receiving BTT. In addition, transfusion of platelets or coagulation factors was also associated with an increased risk of mortality. Blood transfusions may be required in children with increased severity of illness and associated internal organ injuries. These confounding factors may be the reasons for the increased risk of mortality in children who received BTT. Unfortunately, the KID database does not collect injury severity. However, we have used APRDRG severity of illness as a surrogate for severity to adjust the risk of mortality in the regression analysis. In bivariate analysis, the presence of associated thoracic and abdominal injury was associated with increased mortality. Hence, we have included thoracic and abdominal injury in the regression model to adjust the risk of mortality

**Table 3. Univariate analysis of mortality among children with traumatic brain injury and mechanical ventilation with and without blood transfusion therapy.** The values are presented as percentages (95% Confidence intervals).

| Variable | Blood transfusion group % (95%, CI) (n = 3,184) | No Blood transfusion group % (95%, CI) (n = 14,014) | OR (CI); p-value |
|---|---|---|---|
| *Gender* | | | |
| Male | 32.8 (30.5–35.3) | 14.5 (13.7–15.4) | 0.66 |
| Female | 29.4 (26.4–32.6) | 13.9 (12.7–15.2) | |
| *Race/Ethnicity* | | | |
| White | 32.4 (29.6–35.4) | 13.7 (12.7–14.6) | P<0.001 |
| Black | 36 (31.5–40.8) | 16.6 (14.9–18.6) | |
| Hispanic | 27.7 (23.6–32.2) | 12.6 (11.1–14.3) | |
| Other | 33 (27–39.7) | 13.9 (11.6–16.5) | |
| *Age groups* | | | |
| Infants | 23.8 (19.7–28.4) | 19.9 (16.9–23.3) | P<0.001 |
| Toddlers | 35.7 (30–41.8) | 17.8 (15.2–20.7) | |
| Early childhood | 21.7 (15.929) | 10.7 (8.5–13.3) | |
| Middle childhood | 23 (18–28.8) | 9.7 (8.1–11.6) | |
| Early adolescence | 34.2 (30.9–37.5) | 14 (13–15.1) | |
| Late adolescence | 38.8 (34.6–43.2) | 15.5 (14.1–16.9) | |
| *Region* | | | |
| Northeast | 31.7 (26.7–37.1) | 12 (10.4–13.9) | P<0.001 |
| Midwest | 32.8 (28.8–36.9) | 14.9 (13.5–16.3) | |
| South | 32.4 (29.5–35.3) | 16.2 (15.1–17.4) | |
| West | 28.6 (24.6–32.8) | 12 (10.8–13.3) | |
| *Insurance payer* | | | |
| Government | 27.9 (25.2–30.8) | 14.2 (13.2–15.3) | P<0.001 |
| Private | 31.5 (28.7–34.5) | 12.8 (11.8–13.8) | |
| Other | 42.9 (37.7–48.3) | 20 (18–22.2) | |
| *Comorbidity/Complications* | | | |
| Thoracic and abdominal injury vs. No thoracic and abdominal injury | 31.7 (2934.4) vs. 31.6 (28.9–34.3) | 17.2 (1618.5) vs. 12.7 (12–13.6) | 1.4(1.3–1.5); <0.001 |
| NAT vs. No NAT | 27 (21.832.8) vs. 32.2 (30.2–34.3) | 25.3 (2130.2) vs. 14 (13.3–14.7) | 1.7 (1.4–2); < 0.001 |
| Septic shock vs. No septic shock | 22.9 (12.837.6) vs. 31.8 (29.9–33.7) | 17.7 (12.3–24.9) vs. 14.3 (13.615) | 1.1 (0.7–1.5); <0.001 |
| Bacterial infection bacterial infection vs. No | 5.9 (3.5–9.7) vs. 34.6 (32.6–36.7) | 4.1 (3–5.7) vs. 15.3 (14.6–16.1) | 0.2 (0.1–0.2); <0.001 |
| CPR/Cardiac arrest CPR/Cardiac arrest vs. No | 84.4 (79.888.1) vs. 23.7 (21.9–25.6) | 74.1 (70.1–77.7) vs. 11.2 (10.6–11.8) | 22.9 (19.7–26.6); <0.001 |
| *Interventions* | | | |
| EEG vs. No EEG | 18.3 (12.625.9) vs. 32.4 (30.5–34.4) | 12.3 (9.316.1) vs. 14.4 (13.7–15.1) | 0.7 (0.6–0.94); 0.01 |
| Platelet transfusion Platelet transfusion vs. No | 52.4 (47.457.3) vs. 27.4 (25.5–29.5) | 31 (20.443.9) vs. 14.3 (13.615) | 5 (4.2–5.9); <0.001 |
| CF Transfusion vs. transfusion No CF | 70.2 (52.983.2) vs. 31 (29.2–33) | 39.3 (15.4–69.7) vs. 14.3 (13.715) | 8.2 (4.8–13.9); <0.001 |
| EVD vs. No EVD | 3.1 (0.8–11.5) vs. 32.4 (30.5–34.4) | 3.4 (1.6–7) vs. 14.6 (13.9–15.3) | 0.2 (0.1–0.3); <0.001 |

BTT = blood transfusion therapy, CPR = Cardiopulmonary resuscitation, CF = Coagulation Factor, CI = Confidence Interval, EEG- Electroencephalogram,

EVD = External Ventricular Drain, NAT = Non accidental trauma, OR = Odds Ratio

**Table 4. Logistic regression analysis showing variables affecting mortality in children with traumatic brain injury who received mechanical ventilation.**

| Reference Variable | OR (95% CI) | p-value |
|---|---|---|
| *Age*: *(Ref*: *Infants)* | | |
| Toddlers | 5(1.17–1.94) | 0.001 |
| Early Childhood | 0.8 (0.6–1.19) | 0.214 |
| Middle Childhood | 0.8 (0.63–1.12) | 0.236 |
| Early Adolescence | 1.4 (1.1–1.76) | 0.005 |
| Late Adolescence | 1.5 (1.19–1.94) | 0.001 |
| *Race*: (Ref: White) | | |
| Black | 1.1 (0.93–1.23) | 0.23 |
| Hispanic | 0.8 (0.71–0.94) | 0.005 |
| Other | 1 (0.86–1.24) | 0.684 |
| *MHI*: *(Ref*: *1$^{st}$ quartile)* | | |
| 2$^{nd}$ quartile | 0.84 (0.74–0.96) | 0.01 |
| 3$^{rd}$ quartile | 0.73 (0.64–0.84) | <0.001 |
| 4$^{th}$ quartile | 0.63 (0.54–0.74) | <0.001 |
| *Insurance payer*: *(Ref*: *Government)* | | |
| Private | 1 (0.94–1.2) | 0.277 |
| Other | 1.7 (1.51–2.03) | <0.001 |
| Admission to children's hospital | 0.67 (0.57–0.78) | <0.001 |
| *APR-DRGs SOI*: *(Ref*: *1)* | 11.2 (1.97–64.15) | 0.006 |
| 2 | 19.5 (3.51–108.55) | 0.001 |
| 3 | 83.5 (15.1–463.1) | <0.001 |
| 4 | | |
| CPR or cardiac arrest *(Ref*: *No CPR/CA)* | 18.92 (15.9–22.52) | <0.001 |
| Non-accidental trauma | 1.88 (0.88–1.47) | 0.325 |
| Thoracic and abdominal injury | 0.85 (0.76–0.94) | 0.003 |
| Blood transfusion | 1.73 (1.53–1.95) | <0.001 |
| Platelet transfusion | 2.08 (1.67–2.58) | <0.001 |
| Coagulation factor transfusion | 3.37 (1.8–6.33) | <0.001 |

APR-DRGs = All Patients Refined Diagnosis Related Groups, CA = Cardiac Arrest, CPR = Cardiopulmonary Resuscitation, CI = Confidence Interval, MHI = Median Household Income, OR = Odds Ratio, SOI = Severity of Illness

due to BTT. The results of the Hosmer and Lemeshow test in our analysis indicate that our data fits well with the model we have used.

In addition to the regression model, we used propensity score matching to minimize the confounding effect of other variables and to define the effect of blood transfusion on mortality. Although randomized control studies are the gold standard to assess the effectiveness of any treatment, they may be challenging to perform in an ICU environment for a variety of logistical and ethical reasons. Nevertheless, observational studies with propensity matching are increasing with the availability of large electronic databases. Across diverse critical care topics, propensity score studies published in high-impact journals produced results that were generally consistent with the findings of randomized clinical trials [13]. After propensity score matching, the mortality in children in the BTT group was significantly higher compared to those who did not receive BTT.

We have used two robust methods, regression analysis and propensity score matching, to adjust for the confounding variables. In both analyses, we found the adjusted risk mortality was higher with BTT. In addition, we have used a large database representing national experience across the practice conditions, children's as well as non-children's hospitals. However, we cannot exclude some other variables, such as emergency services response times and transport times which could influence mortality as well as BTT frequencies. Due to the limitations of the database, we could not adjust for such confounding variables.

Similar to our study, in a retrospective database review of two pediatric trauma centers, Acker et al. reported higher mortality with blood product transfusion [11]. In our study, we compared children with severe TBI and BTT with those not receiving BTT and did not include patients with any other blood product transfusion. Acker et al. had mild, moderate, and severe TBI in their cohort, which may explain the lower mortality rates among both groups of children compared to our cohort of severe TBI [11]. Blood product transfusions in children with trauma other than TBI were also associated with poor outcomes. In a prospective study, Orlia-guet et al. described outcome predictors in children with severe trauma and reported an increased mortality rate in children with an emergency blood transfusion of more than 20 mL/kg [14]. A single-center retrospective study with a smaller sample size reported no association between mortality with blood transfusion and hemoglobin level in children with severe TBI [15]. In this study, the factors that were significantly associated with mortality after regression analysis were the presence of abusive head trauma, higher Pediatric Risk of Mortality (PRISM) score, and lower GCS after admission. Our study adjusted for most variables associated with increased mortality in TBI with regression analysis and propensity score-matched analysis. Due to the nature of the KID database, we could not adjust for the PRISM score or GCS.

In general, 9–50% of children admitted to the PICU receive at least one PRBC transfusion [16, 17]. In our study, blood transfusion was received in 18.5% of children with severe TBI, and most of them were in the adolescent age group followed by infants, and they are likely to be female and white. Other studies reported a 10 to 44% incidence of PRBC transfusion in pediatric TBI and mainly in the younger age group [11, 15]. Adolescents constituted a higher proportion of children with severe TBI in our cohort, and it could be explained due to their risk-taking behavior, contact sports, and motor vehicle injuries [18].

In our cohort, thoracic and abdominal injury was the most common associated injury and these children more often received blood transfusions. It is expected that children who have thoracic and abdominal injuries are likely to have increased blood loss and require frequent blood transfusions. The mortality rate was higher with the presence of thoracic and abdominal injury in the BTT group in the univariate analysis. However, after adjusting for APRDRG severity of illness and other variables the mortality rate was lower with thoracic and abdominal injuries. It is beyond the scope of this study to explore the reasons for this finding. The severity of thoraco-abdominal injuries could not be determined from the database. Many patients could have had relatively milder thoracic and abdominal injuries which could explain the lower mortality rate associated with thoracic and abdominal injury after adjusting for other confounding variables.

We acknowledge several limitations of our study. The data analysis from an administrative discharge database has its inherent limitations. The data extraction depended on ICD codes, which may or may not have been documented consistently. There was no validated description of TBI severity, such as GCS score, in the KID database. There were no laboratory data, especially hemoglobin levels and PRBC transfusion thresholds were unknown. Physiologic illness severity scores such as the Pediatric Index of Mortality (PIM) or PRISM are used for quantification of physiological status using predetermined physiologic variables to facilitate accurate mortality risk estimation. These scores were not used for risk adjustment in our study. Instead,

we have used APRDRG severity of illness for risk adjustment. However, APRDRG is found to be a good risk-adjusting model for mortality [19]. Another limitation was the lack of a GCS score and use of invasive mechanical ventilation as a surrogate for TBI severity. Our study assessed the national experience of blood transfusion on mortality in mechanically ventilated TBI children compared to previous studies. The study results cannot be extrapolated to children with mild to moderate TBI. Although we have adjusted for the effect of blood product transfusions, we could not assess the effect of TBI-induced coagulopathy, which may be associated with poor outcomes [20, 21]. We cannot deduce a causal relationship between BTT and mortality from our study.

## Conclusions

Based on the analysis of a large administrative national database, we conclude that blood transfusion therapy in children with TBI requiring mechanical ventilation is associated with increased mortality. Although, our study findings favor a restrictive transfusion strategy, additional studies are warranted to support restrictive transfusion practices to mitigate the adverse effects of blood transfusions in children with TBI.

## Acknowledgments

Preliminary results were presented as a poster at the Society of Critical Care Medicine 45th Annual meeting in February 2016.

## Author Contributions

**Conceptualization:** Madhuradhar Chegondi, Balagangadhar R. Totapally.

**Data curation:** Madhuradhar Chegondi, Jose F. Hernandez Rivera, Balagangadhar R. Totapally.

**Formal analysis:** Madhuradhar Chegondi, Fuad Alkhoury, Balagangadhar R. Totapally.

**Investigation:** Balagangadhar R. Totapally.

**Project administration:** Balagangadhar R. Totapally.

**Resources:** Balagangadhar R. Totapally.

**Software:** Balagangadhar R. Totapally.

**Supervision:** Balagangadhar R. Totapally.

**Validation:** Fuad Alkhoury, Balagangadhar R. Totapally.

**Writing – original draft:** Madhuradhar Chegondi, Jose F. Hernandez Rivera.

**Writing – review & editing:** Madhuradhar Chegondi, Jose F. Hernandez Rivera, Fuad Alkhoury, Balagangadhar R. Totapally.

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
