## [Decision Letter · Decision Letter 0]

7 Sep 2022

PONE-D-22-14622The need for blood transfusion therapy is associated with increased mortality in children with traumatic brain injuryPLOS ONE

Dear Dr. Chegondi,

Thank you for submitting your manuscript to PLOS ONE. After careful consideration, we feel that it has merit but does not fully meet PLOS ONE’s publication criteria as it currently stands. Therefore, we invite you to submit a revised version of the manuscript that addresses the points raised during the review process.

Two external reviewers have now evaluated your submission. They have identified a number of concerns that need to be carefully addressed in a revision of the manuscript. Please respond to all of the points they have raised, paying particular attention to their requests for clarification regarding aspects of the methods and data interpretation.

We look forward to receiving your revised manuscript.

Kind regards,

Jamie Males

Editorial Office

PLOS ONE

Journal Requirements:

Reviewers' comments:

Reviewer's Responses to Questions

**Comments to the Author**

1. Is the manuscript technically sound, and do the data support the conclusions?

Reviewer #1: Yes

Reviewer #2: Partly

2. Has the statistical analysis been performed appropriately and rigorously? 

Reviewer #1: Yes

Reviewer #2: Yes

3. Have the authors made all data underlying the findings in their manuscript fully available?

Reviewer #1: Yes

Reviewer #2: Yes

4. Is the manuscript presented in an intelligible fashion and written in standard English?

Reviewer #1: Yes

Reviewer #2: Yes

5. Review Comments to the Author

Reviewer #1: The authors present data from the KID database evaluating the relationship between traumatic brain injury and blood product transfusion. They found that among children with severe TBI (defined by need for mechanical ventilation for >96 hours), blood transfusion was associated with higher risk of mortality.

Methods

1. What is meant by “all data were weight according to HCUP recommendations before analysis to calculate national estimates.” Can the authors add a bit more robust description of HCUP and how the KID database is created? Does the KID database include non children’s hospitals? One of the variable in the logistic regression model is admission to a children’s hospital. It would help provide clarity I the description of the database were more thorough

Results

1. What is meant by internal injury? Does this include blunt liver and spleen injury? Any intraabdominal or intrathoracic injury? This is an odd term to describe trauma.

2. How were total charges defined?

3. Table 3 – it seems like the mortality rate for children who received a blood transfusion and had sepsis was lower than the kids in the transfusion group who did not have sepsis (22.9 vs 31.8%) is this correct? Same thing with bacterial infection, mortality of 5.9% in the no infection group compared to 34.6% in the infection group.

4. Subset analysis of BTT patients – the authors report a median length of stay of 21.1 vs 3.2 days (range 2.8-3.6 days) for survivors vs those who died. I thought the inclusion criteria for the study were mechanical ventilation for a minimum of 96 hours. How can the median length of stay be less than the minimum hours of ventilation needed for study inclusion?

5. The description of the model is difficult to interpret. As a clinician, what does it mean that the model correctly classified 86.4% of cases or the model explained 33% of variance? How can a clinician use this information.

Discussion

1. I was expecting the discussion to spend more time walking the reader through the logistic regression but the discussion really feels entirely separate from the results that are presented. The manuscript is not very cohesive. Instead the discussion seems more of a series of statements about the published literature instead of a thoughtful critique of how a practicing surgeon can understand and use the data presented in the manuscript.

2. Why bring up trauma induce coagulopathy? It comes as a non sequitur in the discussion.

3. Please define PRISM

Reviewer #2: Dear authors, congratulations on your work. I have some minor points to discuss:

- In the logistic regression the presence of internal injuries had a protective effect on mortality ... I found it a bit odd. Was there any interaction between variables that could explain these results? Or how do you interpret it ?

- In the discussion you write: "Although the overall sepsis incidence is lower in pediatric TBI [22], blood transfusion is known to be associated with an increased risk of bacterial infections and other adverse effects [23].Therefore, a higher incidence of bacterial infection and septic shock among the BTT group in our

study is not surprising."This conclusion can only be drawn in the infection/sepsis occurred after blood transfusion and based on the results presented it is not clear for the reader if this is in fact the case.

- In the conclusion you write that "Although, our study findings favor restrictive transfusion strategy, additional studies are warranted to support restrictive transfusion practices to mitigate the adverse effects of

blood transfusions in children with TBI." I agree that restrictive approach appears to be better than a liberal strategy, however, you did not provide information regarding the trigger used to transfuse the patients (perhaps because this data was not available). If the data is available it would be interesting to show how many actually were transfused only when Hb< 7g/dl and compare the mortality rate between patients with a restricted and those with a liberal policy.

6. PLOS authors have the option to publish the peer review history of their article (what does this mean?). If published, this will include your full peer review and any attached files.

Reviewer #1: **Yes: **Shannon Acker MD

Reviewer #2: **Yes: **Elisa Gouvêa Bogossian

---

## [Author Response · Author response to Decision Letter 0]

10 Oct 2022

Reviewer #1: The authors present data from the KID database evaluating the relationship between traumatic brain injury and blood product transfusion. They found that among children with severe TBI (defined by need for mechanical ventilation for >96 hours), blood transfusion was associated with higher risk of mortality.

Methods

Comment1. What is meant by “all data were weight according to HCUP recommendations before analysis to calculate national estimates.” Can the authors add a bit more robust description of HCUP and how the KID database is created? Does the KID database include non children’s hospitals? One of the variable in the logistic regression model is admission to a children’s hospital. It would help provide clarity I the description of the database were more thorough

Response: We apologize for not being clear. We have added the following paragraph in the methods section to describe the HCUP and KID database

The KID contains hospital discharge information from U.S. hospitals that participate in HCUP. Participating institutions are defined as short-term, non-federal, general, and specialty hospitals, excluding rehabilitation hospitals. The KID incorporates information on utilization, access, charges, and outcomes related to children’s medical conditions and the procedures undertaken to treat them. Fully anonymized clinical and resource use information found in a typical discharge abstract is included. Normal newborns were sampled at a rate of 10%, while complicated newborns and other pediatric discharges (age 20 years or less at the time of admission) were sampled at a rate of 80%. Sampling weight is provided for each record so that national; estimates can be calculated. 

Results

Comment 2: 1. What is meant by internal injury? Does this include blunt liver and spleen injury? Any intraabdominal or intrathoracic injury? This is an odd term to describe trauma.

Response: Thank you for raising this important question.

The Clinical Classification Software Diagnoses (CCS) and Clinical Classification Software for Procedures (CCS-PCS) codes identify diagnostic and procedure groups, respectively. CCS and CCS-PCS were created by HCUP to merge multiple individual ICD-10 CM or ICD-10 PCS codes into a smaller number of clinically meaningful categories for a more uniform and standardized coding system. ((HCUP) AfHRaQHCaUP. Introduction to the HCUP KIDS’ inpatient database. www.hcup-us.ahrq.gov: 2019). We have used CCS code (234) to identify the internal injury. The CCS code 234 includes multiple trauma codes for thoracic and abdominal injuries.

We have added this paragraph in the methods section. 

 We modified the internal injury as “thoracic and abdominal injury” throughout the manuscript and in the tables. 

Comment 3: 2. How were total charges defined?

Response: As per HCUP, total charges include total hospital charges excluding professional charges and non-covered charges.

Comment 4: 3. Table 3 – it seems like the mortality rate for children who received a blood transfusion and had sepsis was lower than the kids in the transfusion group who did not have sepsis (22.9 vs 31.8%) is this correct? Same thing with bacterial infection, mortality of 5.9% in the no infection group compared to 34.6% in the infection group.

Response: Thank you for the comment. 

We rechecked our data. Yes, the mortality rate among children who received BTT and had bacterial infection and sepsis were lower compared to children who didn’t have either of them. This may be likely related to the severity of the TBI and the length of hospital stay. It appears that those who survived and stayed longer in the hospital had a higher risk of having sepsis and bacterial infections. 

Comment 5: 4. Subset analysis of BTT patients – the authors report a median length of stay of 21.1 vs 3.2 days (range 2.8-3.6 days) for survivors vs those who died. I thought the inclusion criteria for the study were mechanical ventilation for a minimum of 96 hours. How can the median length of stay be less than the minimum hours of ventilation needed for study inclusion?

Response: We want to thank the reviewer for pointing out this important discrepancy in our manuscript. We completely agree with the reviewer's statement. 

We have reviewed our data for the selection of cases. We have used CCS procedure code 216 for ventilation (which includes all codes for ventilation and intubation). Intubation and ventilation for operating room procedures are not included. We have updated our methods section and other relevant areas and added the description of CCS terminology. We apologize for our oversight for not updating the methods. We have verified the data for the accuracy of values presented in the manuscript. The total number of children in the KID database 2012 and 2016 with intracranial injury and ventilation (CCS code 216) are 17,199. 

Comment 6: 5. The description of the model is difficult to interpret. As a clinician, what does it mean that the model correctly classified 86.4% of cases or the model explained 33% of variance? How can a clinician use this information.

Response: Thank you for the comment. 

The description gives information about the robustness of the model and is useful for comparing the model with a new dataset in future research work. The 33% variance means the prediction of outcome is better by 33% compared to a flip of a coin. We are more than willing to delete the description if it is unhelpful information in a clinical manuscript.

Discussion

Comment 6: 1. I was expecting the discussion to spend more time walking the reader through the logistic regression but the discussion really feels entirely separate from the results that are presented. The manuscript is not very cohesive. Instead the discussion seems more of a series of statements about the published literature instead of a thoughtful critique of how a practicing surgeon can understand and use the data presented in the manuscript.

Response:

Thank you for the suggestion. In response to the reviewer’s suggestion, we have revised the entire discussion section. Focused on the regression and propensity score matching analyses and deleted other discussion not directly related to mortality and blood transfusion therapy.

Comment 7: 2. Why bring up trauma induce coagulopathy? It comes as a non sequitur in the discussion.

Response: We apologize for not being clear on trauma-induced coagulopathy.

Although our study objective was to evaluate the effect of PRBC in TBI, a significant number of patients in our cohort also received other blood products such as platelets, FFP, and cryoprecipitate. Which is likely explained by trauma-induced coagulopathy (TIC). By discussing the TIC, we are trying to highlight the severity of TBI in our cohort. 

Per your suggestion, we deleted this paragraph in the discussion. 

Comment 8: 3. Please define PRISM

Response: We expanded the PRISM acronym as follows

“Pediatric risk of mortality”.

Reviewer #2: Dear authors, congratulations on your work. I have some minor points to discuss:

Comment 1: - In the logistic regression the presence of internal injuries had a protective effect on mortality ... I found it a bit odd. Was there any interaction between variables that could explain these results? Or how do you interpret it ?

Response: Thank you for the pertinent comment. 

The adjusted mortality in our model was surprisingly lower when internal injury was present. A major interaction was with APRDRG severity of illness (SOI). When we ran the model after the exclusion of APRDRG severity of illness, the adjusted odd ratio of mortality with internal injury was higher at 1.21 (95% CI: 1.1-1.3, p<0.001). The adjusted mortality with blood transfusion increased to 2.13 (95% CI: 1.9-2.4, p<0.001). However, when we tested for multicollinearity, there was no collinearity between internal injury, APRGRG SOI, and blood transfusion. 

Comment 2: - In the discussion you write: "Although the overall sepsis incidence is lower in pediatric TBI [22], blood transfusion is known to be associated with an increased risk of bacterial infections and other adverse effects [23].Therefore, a higher incidence of bacterial infection and septic shock among the BTT group in our study is not surprising. "This conclusion can only be drawn in the infection/sepsis occurred after blood transfusion and based on the results presented it is not clear for the reader if this is in fact the case.

Response: Thank you so much for your comment. 

Per your suggestion, given the lack of clarity we deleted this in the discussion. 

Comment 3: - In the conclusion you write that "Although, our study findings favor restrictive transfusion strategy, additional studies are warranted to support restrictive transfusion practices to mitigate the adverse effects of blood transfusions in children with TBI." I agree that restrictive approach appears to be better than a liberal strategy, however, you did not provide information regarding the trigger used to transfuse the patients (perhaps because this data was not available). If the data is available it would be interesting to show how many actually were transfused only when Hb< 7g/dl and compare the mortality rate between patients with a restricted and those with a liberal policy.

Response: Thank you so much for the valuable input. 

As stated by the reviewer, the KID database does not include laboratory values. Hence, we can not determine the optimal or safe transfusion threshold from our study.

---

## [Decision Letter · Decision Letter 1]

26 Oct 2022

PONE-D-22-14622R1The need for blood transfusion therapy is associated with increased mortality in children with traumatic brain injuryPLOS ONE

Dear Dr. Chegondi,

Thank you for submitting your manuscript to PLOS ONE. After careful consideration, we feel that it has merit but does not fully meet PLOS ONE’s publication criteria as it currently stands. Therefore, we invite you to submit a revised version of the manuscript that addresses the points raised during the review process.

Dear Dr. Chegondi, 

thank you for your important work on blood transfusion in children with traumatic brain injury. As a newly assigned editor to you, I have a few comments and criticisms.

I miss an introduction/explanation of the TBI-MV group. What are you trying to show with this? In this group, there is no differentiation between transfusion and no transfusion. 

Choosing 96 hours of ventilation as a surrogate for severe TBI seems arbitrary to me. What is the ratio for this?

The tables or the contents of the table are only partially clear to me and need to be presented and explained more clearly. For example, Table 1, which is supposed to represent demographics, shows a value of 36.5 (34.6- 38.5) in the transfusion group for female gender. What does this mean? I would expect an absolute number and a real value that reflects the proportion of the group. Analogously, all other dichotomous values in the tables 1-3. 

The column "TBI-MV" creates confusion and little added value, since column 4 gives odds ratios and p-values that presumably come from comparing the "blood transfusion" and "no blood-transfusion" groups. 

Furthermore, it is not clear to which row or subgroup the p-values refer.

I ask you to revise the manuscript again and make it more intuitively understandable for the reader. In particular, the content of the table should be easy to grasp and understand.

We look forward to receiving your revised manuscript.

Kind regards,

Alexander Wolf

Academic Editor

PLOS ONE

Reviewers' comments:

Reviewer's Responses to Questions

**Comments to the Author**

1. If the authors have adequately addressed your comments raised in a previous round of review and you feel that this manuscript is now acceptable for publication, you may indicate that here to bypass the “Comments to the Author” section, enter your conflict of interest statement in the “Confidential to Editor” section, and submit your "Accept" recommendation.

Reviewer #1: (No Response)

Reviewer #2: All comments have been addressed

2. Is the manuscript technically sound, and do the data support the conclusions?

Reviewer #1: Yes

Reviewer #2: Yes

3. Has the statistical analysis been performed appropriately and rigorously? 

Reviewer #1: Yes

Reviewer #2: Yes

4. Have the authors made all data underlying the findings in their manuscript fully available?

Reviewer #1: Yes

Reviewer #2: Yes

5. Is the manuscript presented in an intelligible fashion and written in standard English?

Reviewer #1: Yes

Reviewer #2: Yes

6. Review Comments to the Author

Reviewer #1: The manuscript is significantly improved and I commend the authors for their efforts. There are still a few issues that have not been addressed. The definition of internal injury needs to be included in the manuscript; practicing surgeons who read this paper will want to know what this term means and why it was used. I did not see PRISM or PIN defined in the manuscript. The description of the regression is improved. I still think it would be helpful for the clinical surgeon if the authors were more explicit about what these statistics mean. What does it mean that the model correctly classified 86% of cases? Is it that the model correctly predicted mortality in 86% of cases? The writing should be so clear that a surgeon with no knowledge of statistics knows exactly how these data impact clinical practice.

Reviewer #2: The authors have answered all my queries. Thank you and I have no further comments.

Congratulations on your work.

7. PLOS authors have the option to publish the peer review history of their article (what does this mean?). If published, this will include your full peer review and any attached files.

Reviewer #1: No

Reviewer #2: **Yes: **Elisa Gouvêa Bogossian

---

## [Author Response · Author response to Decision Letter 1]

22 Nov 2022

Academic Editor

Comment 1: I miss an introduction/explanation of the TBI-MV group. What are you trying to show with this? In this group, there is no differentiation between transfusion and no transfusion. 

Response: We apologize for not being clear with the explanation. In the KID database GCS score was not available to classify the severity of TBI. Hence, we chose mechanical ventilation as a surrogate marker for severity, and we classified all patients with TBI who were on mechanical ventilator were considered as severe TBI. We have included this explanation in the methods section. 

Comment 2: Choosing 96 hours of ventilation as a surrogate for severe TBI seems arbitrary to me. What is the ratio for this?

Response: In the last revised version we deleted 96 hours cut off and highlighted in the track changes. In the methods section we have mentioned as following

“We included patients who were discharged from hospitals with a diagnosis of TBI with ages between 1 month to less than 21 years who were mechanically ventilated.”

Comment 3: The tables or the contents of the table are only partially clear to me and need to be presented and explained more clearly.

For example, Table 1, which is supposed to represent demographics, shows a value of 36.5 (34.6- 38.5) in the transfusion group for female gender. What does this mean? I would expect an absolute number and a real value that reflects the proportion of the group. Analogously, all other dichotomous values in the tables 1-3. 

Response: Thank you for the opportunity to improve the clarity of our presentation. In response to the academic editor’s comments, we have revised the tables in the new version of the manuscript. The number of patients in each group is included in the column head and values for each variable are presented as % and 95% CI. We have added the following sentence in the table titles for Tables 1-3:

“The values are presented as percentages (95% Confidence intervals)”

We chose to present the values as % for easy visualization of proportions. However, we are more than happy to change the values to absolute numbers.

Comment 4: The column "TBI-MV" creates confusion and little added value, since column 4 gives odds ratios and p-values that presumably come from comparing the "blood transfusion" and "no blood-transfusion" groups. 

Furthermore, it is not clear to which row or subgroup the p-values refer.

Response: Thank you for the suggestion. We have deleted the TBI-MV column in Tables 1-3.

Reviewer #1: 

Comment 1: The manuscript is significantly improved and I commend the authors for their efforts. There are still a few issues that have not been addressed. The definition of internal injury needs to be included in the manuscript; practicing surgeons who read this paper will want to know what this term means and why it was used.

Response: Thank you so much for your feedback. In the methods section, we have included the definition of internal injury as follows:

“Thoracic and abdominal injury resulting from blunt trauma”

Comment 2: I did not see PRISM or PIN defined in the manuscript. 

Response: We expanded the PRISM acronym as “Pediatric Risk of Mortality”. In response to the reviewer’s comment, we have added the following sentence in the limitations section.

“Physiologic illness severity scores such as the Pediatric Index of Mortality (PIM) or PRISM are used for quantification of physiological status using predetermined physiologic variables to facilitate accurate mortality risk estimation. These scores were not used for risk adjustment in our study.”

Comment 3: The description of the regression is improved. I still think it would be helpful for the clinical surgeon if the authors were more explicit about what these statistics mean. What does it mean that the model correctly classified 86% of cases? Is it that the model correctly predicted mortality in 86% of cases? The writing should be so clear that a surgeon with no knowledge of statistics knows exactly how these data impact clinical practice.

Response: Thank you for the comment. We have added the following sentence in the results section of the revised manuscript.

“The model has correctly predicted survival or death in 86.4% of cases.”

Reviewer #2: 

Comment: The authors have answered all my queries. Thank you and I have no further comments.

Response: Thank you so much.

---

## [Editor Report · Decision Letter 2]

13 Dec 2022

The need for blood transfusion therapy is associated with increased mortality in children with traumatic brain injury

PONE-D-22-14622R2

Dear Dr. Chegondi,

We’re pleased to inform you that your manuscript has been judged scientifically suitable for publication and will be formally accepted for publication once it meets all outstanding technical requirements.

Kind regards,

Alexander Wolf

Academic Editor

PLOS ONE
---

## [Editor Report · Acceptance letter]

27 Dec 2022

PONE-D-22-14622R2 

The need for blood transfusion therapy is associated with increased mortality in children with traumatic brain injury 

Dear Dr. Chegondi:

I'm pleased to inform you that your manuscript has been deemed suitable for publication in PLOS ONE. Congratulations! Your manuscript is now with our production department. 

Kind regards, 

on behalf of

Dr. Alexander Wolf 

Academic Editor

PLOS ONE